# A Multi-System Approach to Investigate Different Forms of Delinquency in Female Adolescents at Risk: Family, School, and Peers

**DOI:** 10.3390/bs13120960

**Published:** 2023-11-22

**Authors:** Jerf W. K. Yeung

**Affiliations:** 1Department of Behavioural and Social Sciences, City University of Hong Kong, Hong Kong, China; ssjerf@gmail.com; 2Graduate School of Human Sciences, Osaka University, 565-0871 Osaka, Japan

**Keywords:** person-in-environment perspective, social systems theory, contextual systems, female adolescents at risk, delinquency

## Abstract

The delinquency of female adolescents at risk has increased conspicuously, much to the concern of human service and public health practitioners. Nevertheless, prior findings of pertinent research have commonly been based on samples of male or gender-mixed participants, especially general youths in the community. These cannot adequately explicate the impacts of different contextual systems on at-risk female adolescents, who are, in fact, more vulnerable to developing delinquency. Based on cross-sectional data collected from a sample of 236 at-risk female adolescents with the help of a large social work organization, the present study investigated and compared the effects of family, school, and peer systems on different forms of delinquency among at-risk female adolescents. The results show that positive family functioning, constructive school experiences, and deviant peer influence were predictive of both self-destructive and other-destructive delinquency in at-risk adolescent girls, respectively, but the effects were significantly different depending on which contextual systems influenced what forms of delinquency. Furthermore, the relationships between positive family functioning and both forms of delinquency were significantly mediated by the effects of constructive school experiences and deviant peer influence, giving support to the distal role of family and the proximal functions of school and peers in adolescence. Implications for policy prevention and interventions to strengthen the contextual supports in different social systems to help female adolescents at risk are discussed.

## 1. Introduction

The concern regarding at-risk female adolescents’ delinquency has drawn attention from researchers, as these adolescent girls are generally more susceptible to the influences of their contextual environment due to the fact that females are generally more sociotropic and relationally linked to their social systems [1,2]. This is consonant with the professions of human services and public health that stress the scrutiny of multiple contextual systems in influencing human development and outcomes, which is in accordance with the person-in-environment perspective [3] and social systems theory [4]. Manifestly, with the help of these multi-system approaches, the generalist practice of human service and public health professionals can successfully generate effective interventions and services to solve the problems and needs of clients who are at the mercy of dysfunctions derived from their contextual living systems. Female adolescents at risk are of particular concern for the aforementioned professions. This is because many of their maladjustments and their pathological development can be attributed to the adverse influence of the contextual systems related to their family, school, and peers [5,6,7]. In the present study, I sought to investigate and compare the effects of family, school, and peer systems on the development of different forms of delinquency in a sample of female adolescents at risk who were clients of frontline social workers.

Apparently, examining and comparing the effects of family, school, and peer systems on the delinquent behaviors of female adolescents at risk are of theoretical and practical importance. First, knowledge on how different contextual systems contribute to the delinquency of female adolescents at risk is insufficient, as prior empirical studies tended to generate their results from samples of male or gender-mixed participants, especially based on general community youths [7,8,9,10]. Second, the female gender is more sociotropic and relationally bound [11,12], hence more susceptible to the influences of family, school, and peer systems. Third, with the rising view of normalizing female delinquency, it is evident that the prevalence of girls’ delinquency increases dramatically and gravely to the extent of pronounced concerns of human service and public health professions [12,13]. Fourth, delinquent adolescent girls involved in the juvenile justice system have been found to have higher exposure to multiple contextual risks and problems simultaneously compared to their male counterparts, which is called the “threshold hypothesis” by researchers [2]. This denotes that female adolescents need to pass a higher threshold of risks to become delinquent. Last and more important, due to the intergenerational transmission perspective, delinquent and maladjusted female adolescents may badly hazard the positive development and well-being of their offspring through prospective maternal psychopathology and harsh parenting [2,12]. Accordingly, it is of particular interest to investigate and determine the effects of family, school, and peer systems on the delinquency of adolescent girls at risk, which may merit early prevention and interventions for this specific youth group before the irreversibility of their problems and pathology becomes apparent.

## 2. Theoretical Framework

Human development and outcomes are inseparable from their living environments; these are called contextual systems or social systems in public health and social science terminology. Consonant with the person-in-environment perspective [3] and social systems theory [4], the contextual or social systems which we inhabit are important sources of socialization, nurturing, and gain of experiences, which in turn shape our personality and behaviors [7,14]. While the person-in-environment perspective highlights the crucial effects of external and social forces on human outcomes [3], social systems theory focuses on the reciprocal relationships between an individual and her or his environments [4]. Both, however, insist that the “individual cannot be understood adequately without reference to the environmental context the individual inhabits [15].” In this study, I attempt to investigate the effects of family, school, and peer systems, which are thought to be most pertinent to youth development, on different forms of delinquency in at-risk female adolescents, e.g., other-destructive and self-destructive delinquency. Moreover, different contextual systems may have different socialization accentuations and, thus, influence the development of youth differently [4,16]. Hence, I also examine whether family, school, and peer systems have significantly different effects on the aforementioned different forms of delinquency in female adolescents at risk. Finally, adolescence is a transitional period for youths to strive for autonomy and independence from family, spend more time in school, and/or interact with peers [10,16,17]. Therefore, another focus of this study is whether the school and peer effects, deemed the most proximate systems to affect youth development in adolescence, would mediate the influences of family, which is considered a comparatively distal system, on the different forms of delinquency in at-risk adolescent girls.

Compared to other contextual systems, family has long been reckoned as the most fundamental and influential contextual system to affect youth development and well-being [8,14,18,19]. Numerous studies have supported the idea that family characteristics of parental warmth, effective discipline, and supportive parent–child relationships—the manifestation of positive family functioning—could deter adolescents from developing maladjustment and delinquency [6,20,21]. In contrast, dysfunctional family socialization, such as conflicting parent–child interactions, hostile parenting, and a lack of family cohesion, is related to higher psychological and behavioral problems in adolescents [7,9]. Adolescent delinquency in this study refers to a series of illegal, socially sanctioned, and immoral acts committed by female adolescents at risk, which are classified as forms of self-destructive and other-destructive behaviors. Other-destructive delinquency concerns involvement in delinquent behaviors that cause harm, devastation, infliction, and loss to others [1,22], e.g., fighting, bullying, gang brawl, and theft. Self-destructive delinquency denotes engagement in delinquent acts that bring hurt, danger, risk, and harm to the actor herself [11,22], e.g., smoking, drinking alcohol, and taking drugs. Family is the basic cardinal socialization agent of child and youth development, and is, therefore, expected to predict both forms of delinquency in female adolescents at risk. However, family generally places more concern on caring relationships, love and intimacy, and self-identity, which are more effective in cultivating adolescents’ self-worth and self-respect and hence causing them to desist from self-harming behaviors [2,23]. For this, it is expected for the family system to have more influence on at-risk female adolescents’ self-destructive delinquency than that of other-destructive delinquency. Hence, the first hypothesis was set accordingly, as follows:

**H1:** 
*Positive family functioning would be negatively related to both self-destructive and other-destructive delinquency of female adolescents at risk, in which a stronger effect of positive family functioning on the form of self-destructive delinquency than that of other-destructive delinquency is expected.*


School is another contextual system that is profoundly influential to adolescent development. In school, adolescents are inculcated with normative values, interpersonal relationships and skills, self-identity and citizenship, and social responsibility through formal socialization and modeling [24,25,26]. Research has found that adolescents who are closely connected and committed to school, specifically the manifestation of constructive school experiences, exhibited fewer deviant and problematic behaviors [24,25,27]. Conversely, if adolescents feel excluded from school, have poor teacher–student relationships, and show less academic enjoyment, which is indicative of negative school experiences, they may engage in more anti-social and delinquent behaviors [17,28]. In addition, school is a formal socialization institution that stresses all-around prosocial and positive development, ranging from self-concept to other-regarding behaviors [25]. It is, thus, anticipated that constructive school experiences would have equal effects on preventing female adolescents at risk from developing both self-destructive and other-destructive delinquency. Therefore, the second hypothesis was set accordingly, as follows:

**H2:** 
*Constructive school experiences would be negatively related to both self-destructive and other-destructive delinquency of female adolescents at risk, in which identical effects of constructive school experiences on self-destructive and other-destructive delinquency are expected.*


Association with peers is another important contextual system for adolescents to develop social skills and establish interpersonal relationships [29,30]. However, the adverse influence of deviant peers on adolescents’ delinquency and adjustment problems is profound and enduring [2,9,31]. Abundant research has proven that involvement with deviant peers apparently contributes to various delinquent behaviors in youths that are net of the effects of family and school influences [6,14,17,32], suggesting that susceptibility to deviant peer influence is a crucial risk factor for adolescent development of delinquency. Nevertheless, according to the deviancy training thesis and opportunity theory [6,9], the influence of deviant peers would more strongly contribute to the form of other-destructive delinquency than that of self-destructive delinquency. This is because engagement in other-destructive delinquency, e.g., fighting, bullying, and theft, requires incitement, rationalization, and “support” from deviant companions [30,33], which is a process called “self-enhancement” [33]. As such, the third hypothesis was set accordingly, as follows:

**H3:** 
*Susceptibility to deviant peer influence would be positively related to both self-destructive and other-destructive delinquency of female adolescents at risk, in which a stronger effect of susceptibility to deviant peer influence on other-destructive delinquency than that of self-destructive delinquency is expected.*


The current study investigated the effects of contextual systems of family, schools, and peers on the self-destructive and other-destructive forms of delinquency in female adolescents at risk, informed by the social systems theory and the ecological model that family is the most fundamental and etiologic socialization agent to shape adolescent behavioral choices and attitudes toward other contextual systems [4,34], e.g., school and peers. Thereby, it is possible that school and peer systems would mediate the effects of family system on different forms of delinquency in at-risk female adolescents. Jang and Thornberry [33] found that the influence of parenting practices on adolescent delinquency was indirect through its impact on the mediators of adolescent self-control, delinquent attitudes, affiliation with delinquent peers, and time spent in criminogenic settings for the association between parenting and delinquency. Pertinent studies have reported that poor family functioning first contributes to adolescents’ adverse school experiences and connections with deviant peers, which then become the mediators for adolescent delinquent behaviors [6,9]. These research findings are consistent with the transitional period of adolescence, in which the influence of family tapers off as adolescents strive for autonomy and independence and instead spend more time at school and interact with peers more frequently, elucidating the distal role of family socialization and the proximal functions of school and peer influences during adolescence [17,30]. For this, the fourth hypothesis was set accordingly, as follows:

**H4:** 
*The effects of positive family functioning on both self-destructive and other-destructive delinquency of female adolescents at risk would be mediated by constructive school experiences and susceptibility to deviant peer influence.*


## 3. Research Method

### 3.1. Sample and Data Source

The present study was a cooperation project with the Hong Kong Young Women’s Christian Association (HKYWCA), with the aim of investigating the situations and problems faced by local adolescent girls at risk and designing pertinent health and social services to cater to their needs. The HKYWCA is one of the oldest and largest social service organizations financially supported by the Hong Kong government, and the organization also manifests itself by emphasizing female and gender-related services. The present study recruited 236 at-risk female adolescents who were clinical cases of frontline social workers in the organization. The data collection procedures were conducted by first introducing the study purpose and contents to pertinent frontline social workers in the organization who served female adolescents at risk. The frontline social workers then helped to contact their at-risk female adolescent clients and seek consent from them and their parental guardians, if needed, to take part in the study. After obtaining consent from the at-risk female adolescents and their parental guardians if needed, questionnaires containing question items regarding family, school, and peer experiences; cognitive and personality traits; and behavioral performance were given to the female adolescents at risk to fill out in the service units of HKYWCA. Generally, the participating at-risk female adolescents took around 20 to 30 min to complete the questionnaire. Finally, a sample of 236 at-risk female adolescents was successfully surveyed for data analysis. The organization of HKYWCA provides various social work services and programs to young people, especially adolescent girls experiencing deprivation and disadvantages. The provided social services and programs include integrated youth centers, school social work, outreach social work, life orientation and career development programs, and family services. The present study collected data from a sample of 236 adolescent girls aged between 11 and 18 years old who were at risk of developing grave maladjustment and antisocial problems, such as premature pregnancy, HIV infection, and serious criminality, if no timely and pertinent health and social service interventions were provided to them. The study was based on a research project titled “Gendered Deviant Behaviors of Female Youths At-Risk: A Multi-System Approach”, which was ethically approved by the ethical review committee of City University of Hong Kong (Reference Number 7005169; 1 September 2018).

### 3.2. Measures

In this study, the measures included self-destructive delinquency and other structural delinquency of at-risk female adolescents treated as outcome variables, and positive family functioning, constructive school experiences, and deviant peer influence were analyzed as the main predictor variables. In addition, the demographic variables of at-risk female adolescents’ age, educational attainment, current working identity, religious beliefs, and civic organization membership were included as covariates to adjust for their confounding effects on the study relationship. The contents of the measurement items for the study variables are listed in Appendix A. 

#### 3.2.1. Delinquency

The self-destructive delinquency of female adolescents at risk was measured by participation in the delinquent acts of truancy, running away, smoking, alcohol drinking, drug taking, unsafe sexual behavior, compensated dating, sex with multiple partners, and gambling in the past three months. Other-destructive delinquency was measured by participation in triadic activities, intimidation, fighting, bullying, gang brawls, theft, trickery, damage to public properties, and stealthy snapping in the past three months. The measurement of these two forms of self-destructive and other-destructive delinquency among at-risk female adolescents was based on recent pertinent studies [11,22]. If the adolescent girl had taken part in any of these delinquent acts in the form of self-destructive or other-destructive delinquency, a score of 1 was assigned. The accumulated count scores thus range from 0 to 9, respectively.

#### 3.2.2. Positive Family Functioning

In this study, six items were used to measure positive family functioning, which characterized quality parent–child relationships, parental concern and care, mutual support, and efficient family communication. These have been used in recent empirical research to measure effective and positive family processes and interactions [21,23]. Example items to be answered by at-risk adolescent girls included “My mother or father will respect and accept my opinions on important issues” and “It is enjoyable for me to get along with my mother or father”. The items were measured on a 4-point scale from almost not (1) to always (4); higher scores denoted better positive family functioning. The Cronbach alpha coefficient was α = 0.890, indicative of very good reliability.

#### 3.2.3. Constructive School Experiences

For constructive school experiences, nine items modeled from existing research measuring positive teacher–student relationships and effective school engagement were used to tap into the constructive school experiences of at-risk female adolescents [35,36]. Example items included “In school, teachers give me adequate opportunities to develop my strengths” and “I enjoy participating in school activities”. The items were measured on a 4-point scale ranging from strongly disagree (1) to strongly agree (4); higher scores denoted more positive school experiences. The internal consistency was α = 0.909, indicative of excellent internal reliability.

#### 3.2.4. Susceptibility to Deviant Peer Influence

To capture the susceptibility of deviant peer influence, five items were used to measure at-risk female adolescents’ inclination toward molding and accepting their deviant peers’ behaviors by referring to existing relevant research [22,31,37]. Example items included “If my friends invite me to take drugs together, I will consider joining them” and “If my friends invite me to bully others, I will consider taking part in it”. The items were measured on a 4-point scale ranging from strongly disagree (1) to strongly agree (4); higher scores represented being more susceptible to the adverse influence of deviant peers. The alpha coefficient was α = 0.900, implying excellent internal consistency.

#### 3.2.5. Background Covariates

The covariates of at-risk female adolescents’ background information were adjusted in the analysis for the purpose of precluding confounding effects, which included age, educational attainment, current working identity, religious beliefs, and membership in civic organizations. Prior research indicates that these background characteristics are influential on the delinquent behaviors of youths [2,9,10,38]. Age was a continuous variable, ranging from 11 to 18 years old. Educational attainment was an ordered categorical variable ranging from junior secondary school or below to undergraduate education. Current working status was a dummy variable (0 = studying; 1 = working). Religious beliefs were constructed as two dummy variables, in which believers of Christianity or other religions, e.g., Buddhism and Taoism, were coded 1 and non-believers were the reference group (0). Membership in any civic organization, such as volunteer or uniform organizations, was coded 1, and otherwise was 0.

### 3.3. Analytical Strategies

Due to the two outcomes of at-risk female adolescents’ self-destructive and other-destructive delinquency, which are count variables and interrelated in nature, r = 0.662, *p* < 0.001, I employed multivariate Poisson regression models to analyze the data. A count outcome variable denotes the numeric responses as positive integers of zero or greater [39], which correspond to the numbers of delinquent acts involving female adolescents at risk in the past three months. Thereby, the assumptions of normality and linear function in OLS linear regression are inapplicable for count data, because a Poisson distribution is discrete and the Poisson mean is always ≥ 0. A Poisson model must exhibit the log outcome rate as a linear function of a set of its predictors [39,40], for example, *log_e_(Y)* = *b_0_* + *b_1_X_1_* + *b_2_X_2_* + …+ *b_k_X_k_*. Poisson regression generates both the beta coefficient and the exponentiated beta value (Exp[b]); the former reveals the magnitude of the effects of positive family functioning, constructive school experiences, and influence of deviant peers on the two forms of delinquency, and the latter presents the percentage of increase in involvement in any delinquent acts with the two forms of delinquency. Thus, both are reported in this study.

To test whether positive family functioning, constructive school experiences, and susceptibility to deviant peer influence have significantly different effects on the forms of self-destructive and other-destructive delinquency, a Wald test of parameter equivalence constraint was performed, in which the regression equations of self-destructive and other-destructive delinquency were pooled in a single model and the regression parameters of self-destructive and other-destructive delinquency were concurrently set as equivalent (*β_self-destructive_* = *β_other-destructive_*) [41]. Rejection of the equivalent constraint hypothesis suggests the effects of positive family functioning, constructive school experiences, and deviant peer influence on the two forms of delinquency are significantly different. In addition, an indirect effect test was used to test whether the effects of positive family functioning on both at-risk female adolescents’ self-destructive and other-destructive delinquency were significantly mediated by constructive school experiences and deviant peer influence [42]. An indirect effect from positive family functioning to self-destructive and/or other-destructive delinquency was modeled through the effects of constructive school experiences and deviant peer influence (and the reverse is true for positive family functioning as a mediator). In modeling the above regression relationships, the background covariates of the female adolescents’ ages, educational attainment, working identity, religious beliefs, and civic organization membership mentioned above were simultaneously adjusted in the analyses. The statistical procedures were performed by M*Plus* 7.11 [43].

## 4. Results

Table 1 shows that the average age of 236 at-risk female adolescents was 15.055, indicating that they were generally in mid-adolescence. The mean score for the at-risk female adolescents’ education attainment was 2.322, revealing that most of them were of senior secondary school education. In addition, 205 female participants were students (*n* = 205), and the remaining 31 participants were working. Among them, 54 and 28 participants were believers in Christianity and other religions, and the remaining 154 female participants were of no religion, respectively. In addition, 61.4% of the sample were members of a civic organization (*n* = 145), and 38.6% of the female participants had not joined any civic organization (*n* = 91). 

Table 2 presents the correlations of the study variables, in which both self-destructive and other-destructive delinquency of at-risk female adolescents were significantly correlated with positive family functioning and constructive school experiences in a negative direction, *r* = −0.149 to −0.483, *p* < 0.05 or 0.001; and significantly correlated with deviant peer influence in a positive direction, *r* = 0.514 and 0.437, *p* < 0.001. In addition, positive family functioning, constructive school experiences, and deviant peer influence were significantly correlated with each other in an expected direction, either positively or negatively.

Table 3 shows the effects of positive family functioning, constructive school experiences, and deviant peer influence on at-risk female adolescents’ self-destructive and other-destructive delinquency using multivariate Poisson regression models. Evidently, the predictors from the family, school, and peer systems were all significantly predictive of female adolescents’ self-destructive and other-destructive delinquency, in which deviant peer influence showed the strongest effects on both the self-destructive and other-destructive delinquency, *β* = 0.397 and 0.443, *p* < 0.001, indicating that a unit increase in deviant peer influence resulted in the increased odds of at-risk female adolescents engaging in self-destructive delinquency by 48.7% and in other-destructive delinquency by 55.7%. Constructive school experiences had the second-strongest effect on at-risk female adolescents’ self-destructive and other-destructive delinquency, *β* = −0.387 and −0.418, *p* < 0.001, in which a unit increase in constructive school experiences reduced the odds of at-risk female adolescents engaging in self-destructive delinquency by 32.1% and in other-destructive delinquency by 38%. Moreover, positive family functioning was also significantly and negatively predictive of at-risk female adolescents’ self-destructive and other-destructive delinquency, *β* = −0.235 and −0.125, *p* < 0.01 and 0.05, explicating that a unit increase in positive family functioning reduced the odds of at-risk female adolescents engaging in self-destructive and other-destructive delinquency by 21% and 11.8%, respectively. As a result, hypotheses 1, 2, and 3 are supported. 

Table 4 displays the results of the Wald parameters equivalence constraint test to compare whether positive family functioning, constructive school experiences, and deviant peer influence had significantly different effects on the self-destructive and other-destructive delinquency of at-risk female adolescents. The results confirmed that positive family functioning had a stronger effect on self-destructive delinquency than on other-destructive delinquency, Δ*β* = 0.110, *p* < 0.001. Inversely, deviant peer influence had a stronger effect on other-destructive delinquency than on self-destructive delinquency, Δ*β* = 0.046, *p* < 0.05. As expected, constructive school experiences did not have significantly different effects on either form of delinquency among at-risk female adolescents, Δ*β* = −0.031, *p* > 0.05. For this, hypotheses 1, 2, and 3 are supported. 

Table 5 presents the total indirect effects of positive family functioning, constructive school experiences, and deviant peer influence on the self-destructive and other-destructive delinquency of at-risk female adolescents. The results showed that constructive school experiences and deviant peer influence significantly mediate the relationships between positive family functioning and at-risk female adolescents’ self-destructive and other-destructive delinquency, *β_ind_* = −0.355 and −0.316, *p* < 0.001. However, positive family functioning was not found to significantly mediate the effects of constructive school experiences on the self-destructive or other-destructive delinquency of at-risk female adolescents (*β_ind_* = −0.029 and 0.046, *p* > 0.05), nor did the effects of deviant peer influence on at-risk female adolescents’ self-destructive and other-destructive delinquency (*β_ind_* = 0.013 and −0.022, *p* > 0.05). Evidently, hypothesis 4 is supported.

## 5. Discussion

As there is a lack of research regarding how different family, school, and peer systems contribute to different forms of delinquency in at-risk female adolescent girls, the present study aimed to investigate the respective and different effects of positive family functioning, constructive school experiences, and deviant peer influences on the self-destructive and other-destructive delinquency of at-risk female adolescents who were cases of frontline social workers. The results confirmed that positive family functioning, constructive school experiences, and deviant peer influence were, respectively, predictive of both forms of delinquency in at-risk female adolescents, indicating that these three contextual systems exert important influences on at-risk female adolescents’ behavioral maladjustment. Thus, when working with delinquent adolescent girls at risk, human service and public health practitioners should contemporaneously consider the impacts of different contextual systems affecting their development and emphasize the different effects in multiple contextual domains [6,25,31,44]. This is congruous with the person-in-environment perspective and ecological framework that human development is susceptible to the social systems which they inhabit [4,34]. Therefore, an important role of human service and public health practitioners is to mediate the influences of different contextual systems and tactically coordinate the necessary resources and supports to maximize the beneficial development of adolescent girls at risk and help them desist from delinquency. For this, human service and public health practitioners should work together with other professionals in educational, medical, and judicial fields across different social systems to collectively cultivate a nurturing and protective socialization environment for the positive development of at-risk female adolescents.

Noteworthy, this study found that different contextual systems had different effects on at-risk female adolescents’ self-destructive and other-destructive delinquency. The results of the Wald parameters equivalence constraint test attested that the influence of the family system was significantly stronger regarding female adolescents’ self-destructive delinquency than other-destructive delinquency, and the reverse was true for the impacts of deviant peer influence, which had a stronger effect on at-risk female adolescents’ other-destructive delinquency than self-oriented delinquency. Moreover, the current study found that the effects of constructive school experiences on both forms of at-risk female adolescents’ delinquency were equivalent. Hence, human service and public health practitioners should note these significantly different effects of family, school, and peer systems on at-risk female adolescents’ development when helping them to solve their personal and behavioral problems. For example, a female adolescent may perform as a good child with self-care in the family realm, but covertly engage in other-destructive behaviors outside of the home [2,13]. Therefore, human service and public health practitioners should, on the one hand, work on the strengths and, on the other hand, remedy the weaknesses of a specific contextual system that contributes to at-risk female adolescents’ self-destructive and other-destructive behavioral maladjustment. 

In addition, this study confirms the distal role of the family system and the proximal functions of the school and peer systems in influencing female adolescents’ delinquency. Results of indirect-effect tests showed that constructive school experiences and deviant peer influence significantly mediated the effects of positive family functioning on both forms of delinquency in female adolescents. However, the reverse mediational relationships were not empirically supported by adopting positive family functioning as a mediator and constructive school experiences and deviant peer influence as the distal predictors. It is true that children in the adolescent stages may strive for autonomy and independence [17,25,45], which indicates the tapering-off effects of family socialization and the magnifying influences of school and peer systems on adolescent development. Nevertheless, this does not signify that family has less of an influence on adolescent development [2,19]. In fact, the effects of family socialization are persistent and steadfast, meaning that instead of exerting direct impacts on child development, families may influence their adolescent children indirectly by swaying their methods of interacting with their schools and peer systems [16,46,47]. Thus, as is consistent with the person-in-environment perspective and social systems theory [3,4], human service and public health practitioners should take note of the dynamic and conditional effects of different contextual systems on female adolescent development [2,5,48,49]. As such, when dealing with different contextual systems in relation to different types of delinquency in female adolescents, more attention should be paid to the dynamic and conditional effects of family, school, and peer systems in relation to their development.

## 6. Conclusions

The present study, to the author’s knowledge, was the first attempt to contemporaneously examine the effects of family, school, and peer systems on at-risk female adolescents’ self-destructive and other-destructive delinquency. It corroborated that these three crucial social systems are, altogether, predictive of the different forms of delinquency with different strengths among this vulnerable population, which can provide valuable empirical evidence for policy and service innovations and interventions to help female adolescents at risk to experience better prosocial development by improving their malfunctioning living environments and creating a nurturing socialization context. Future research should further explore other contextual systems concomitantly in relation to youths’ psychological and behavioral development, e.g., neighborhood and religion. Furthermore, a research lens should focus on how these different contextual systems interact with each other, as well as how different adolescent personality factors, e.g., resilience, align with different contextual systems to contribute to adolescent development. In fact, there are some limitations of the current study that should be rectified in the future. First, the results of this study were only based on a convenient sample of adolescent girls at risk who were recruited by a large NGO in Hong Kong. Future research should invite more NGOs in different geographical areas that provide interventions and services for youths to help increase the sample size and sample diversity for the purpose of enhancing external validity. In addition, cross-nation samples of at-risk female adolescents are suggested for future research, which may help to promote generalizability and empirical comparison. Third, the data of the current study were collected using a cross-sectional design, and future research should use longitudinal research data to trace the development of and changes in at-risk female adolescents’ psychological and behavioral development, as well as how different contextual systems affect development and changes. Lastly, a longitudinal design with a multi-informant approach to data collection is suggested for future studies, which can help to not only verify the causal validity of the study relationships between different contextual systems and at-risk female adolescents’ different forms of delinquency, but also confirm the results from different perspectives in order to obtain a more comprehensive picture of at-risk female adolescent development.

## Figures and Tables

**Table 1 behavsci-13-00960-t001:** Descriptive statistics of at-risk female adolescents’ background characteristics (*n* = 236).

Background Covariates	Mean (Frequency)	SD	Range
Age	15.055	1.753	11 to 18
Educational attainment	2.322	0.581	1 to 4
(1) Junior secondary school or below	0.047 (11)		0, 1
(2) Senior secondary school	0.597 (141)		0, 1
(3) Associate degree/diploma	0.343 (81)		0, 1
(4) College degree or above	0.013 (3)		0, 1
Current working identity			
(1) Studying	0.869 (205)		0, 1
(2) Working	0.131 (31)		0, 1
Religious belief			
(1) Christianity	0.229 (54)		0, 1
(2) Other religion	0.119 (28)		0, 1
(3) No religion	0.652 (154)		0, 1
Civic organization membership			
(1) Yes	0.614 (145)		0, 1
(2) No	0.386 (91)		0, 1

Note. Educational attainment indicates the current studying or graduation level of at-risk female adolescents. Civic organization membership refers to at-risk female adolescents joining any volunteering, uniform, or social service organization as a member rather than a client.

**Table 2 behavsci-13-00960-t002:** Correlations of the study variables.

		1	2	3	4	5
1	Self-destructive delinquency					
2	Other-destructive delinquency	0.662 ***				
3	Positive family functioning	−0.309 ***	−0.149 *			
4	Constructive school experiences	00.483 ***	−0.413 ***	0.403 ***		
5	Deviant peer influence	0.514 ***	0.437 ***	−0.310 ***	−0.393 ***	

* *p* < 0.05; < 0.01; *** *p* < 0.001.

**Table 3 behavsci-13-00960-t003:** Multivariate Poisson regressions predicting at-risk female adolescents’ self-destructive and other-destructive delinquency.

	Outcomes	Self-Destructive		Other-Destructive	
Predictors		*β*	*OR*	*β*	*OR*
Positive family functioning		−0.235 **	0.790	−0.125 *	0.882
Constructive school experiences		−0.387 ***	0.679	−0.418 ***	0.620
Deviant peers influence		0.397 ***	1.487	0.443 ***	1.557

Note. Covariates of at-risk female adolescents’ age, educational attainment, current working identity, religious beliefs, and civic organization membership were adjusted. OR = odds ratio. * *p* < 0.05; ** *p* < 0.01; *** *p* < 0.001.

**Table 4 behavsci-13-00960-t004:** Results of Wald parameters equivalence constraint testing the effects on at-risk female adolescents’ self-destructive and other-destructive delinquency.

	Outcomes	Self-Destructive	Other-Destructive	Difference in Beta	*Wald X* ^2^
Predictors		*β*	*β*	*Δβ*	
Positive family functioning		−0.235 **	−0.125 *	0.110	9.526 ***
Constructive school experiences		−0.387 ***	−0.418 ***	−0.031	2.399
Deviant peers influence		0.397 ***	0.443 ***	−0.046	4.489 *

Note. Covariates of at-risk female adolescents’ age, educational attainment, current working identity, religious beliefs, and civic organization membership were adjusted. Difference in beta (Δ*β*) = *β_self-destructive_* − *β_other-destructive_*. * *p* < 0.05; ** *p* < 0.01; *** *p* < 0.001.

**Table 5 behavsci-13-00960-t005:** Total indirect effects of positive family functioning, constructive school experiences, and deviant peer influence on at-risk female adolescents’ self-destructive and other-destructive delinquency ^a^.

	Outcomes	Self-Destructive		Other-Destructive	
Predictors		*β_ind_*	*t-Value*	*β_ind_*	*t-Value*
Positive family functioning ^b^		−0.355	−5.102 ***	−0.316	−4.856 ***
Positive school experiences ^c^		−0.029	−0.821	0.046	1.270
Deviant peers influence ^d^		0.013	0.782	−0.022	−1.139

Note. ^a^ The indirect effect tests, adjusted for the effects of at-risk female adolescents’ age, educational attainment, current working identity, religious beliefs, and civic organization membership as covariates. ^b^ Indirect effects of positive family functioning on at-risk female adolescents’ self-destructive and other-destructive delinquency were tested through the paths of constructive school experiences and deviant peer influence. ^c^ Indirect effects of constructive school experiences on at-risk female adolescents’ self-destructive and other-destructive delinquency were tested through the path of positive family functioning while adjusting for the effects of deviant peer influence. ^d^ Indirect effects of deviant peer influence on at-risk female adolescents’ self-destructive and other-destructive delinquency were tested through the path of positive family functioning while adjusting for the effects of positive school experiences. *** *p* < 0.001.

## Data Availability

The data are not publicly available due to protection of the at-risk female adolescents’ confidentiality, and unidentifiable data are only available on reasonable request due to restrictions and ethical consideration.

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
