# Peer review of "A Multi-System Approach to Investigate Different Forms of Delinquency in Female Adolescents at Risk: Family, School, and Peers"

_behavsci, 2023, doi:10.3390/bs13120960_

Round 1

Reviewer 1 Report

Comments and Suggestions for Authors

I would like to thank the Editor for the opportunity to review this study and I am flattered to be able to provide my contribution. This is undoubtedly an interesting paper, but the authors should pay attention to a few points:

(a) In the sample section, more sample information should be added in a table. For example, the information used as covariate variables (section 3.2.5) and the information in the first paragraph of the results section could easily be included in this table.

b) There are some inconsistencies regarding the age range of the participants. On page 4 it is stated that it ranges from 11 to 18 years of age, while on page 6 it is stated that it ranges from 12 to 18 years of age.

c) In the section on instruments, the following aspects should be clarified:

a.        The variables included in the delinquency section (section 3.2.1) should be clarified as to where these data are collected. I think I understand that they should be collected from the social workers' records, but it is not clear.

b.        Regarding the other measures (sections 3.2.2, 3.2.3, 3.2.4), since they are non-standardized instruments, the authors should include all the items of each of the measures as an appendix so that the study can be replicated. In addition, the description of the measures should specify that they were completed by the adolescents, as well as in what form they were completed (e.g., self-report, as an interview, etc.).

d) The discussion could be enriched a little. Adding information on other papers that agree or disagree with the results found would greatly enrich the work.

Author Response

For Reviewer 1

I would like to thank the Editor for the opportunity to review this study and I am flattered to be able to provide my contribution. This is undoubtedly an interesting paper, but the authors should pay attention to a few points:

(a) In the sample section, more sample information should be added in a table. For example, the information used as covariate variables (section 3.2.5) and the information in the first paragraph of the results section could easily be included in this table.

Reply: Now more information regarding the procedure of data collection, descriptive statistics of the at-risk female adolescents, and the contents of the measures are added in the parts of ‘3.1 Sample and data source’, ‘3.2 Measures’, ‘4. Results’, and ‘Appendix’ of revised manuscript.

In 3.1 Sample and data source, for the data collection procedure it is written:

“The present study was a cooperation project with the Hong Kong Young Women’s Christian Association (HKYWCA) with the aim of investigating the situations and problems faced by local adolescent girls at risk and designing pertinent health and social services to cater for their needs. HKYWCA is one of the oldest and largest social service organizations financially supported by the Hong Kong government, and the organization also manifests itself by emphasizing female and gender-related services. The present study recruited 236 at-risk female adolescents who were the clinical cases of frontline social workers in the organization. The data collection procedures were conducted by first introducing the study purpose and contents to pertinent frontline social workers in the organization who serve female adolescents at risk; and the frontline social workers then helped to contact their at-risk female adolescent clients and seek consent from them and their parental guardians if needed to take part in the study. After obtaining consents from female adolescents at risk and their parental guardians if needed, a questionnaire containing the question items regarding family, school, and peer experiences, cognitive and personality traits, and behavioral performance was given to the consented female adolescents at risk to fill up in the service units of HKYWCA. Generally, the participating at-risk female adolescents needed to take around 20 to 30 minutes to complete the questionnaire. Finally, a sample of 236 at-risk female adolescents was successfully surveyed for data analysis.” (lines 941-959)

In 3.2 Measures, for the more details of the study variables being measured, it is written:

“3.2. Measures

In this study, the measures include self-destructive delinquency and other-structural delinquency of at-risk female adolescents that are treated as the outcome variables, and positive family functioning, constructive school experiences, and deviant peer influence are analyzed as the main predictor variables, along with adjusting for the effects of demographic covariates of at-risk female adolescentsage, educational attainment, current working identity, religious beliefs, and civic organization memberships. The contents of the measurement items for the study variables are listed in the Appendix.

3.2.1. Delinquency

Self-destructive delinquency of female adolescents at risk was measured by participation in the delinquent acts of truancy, runaway, smoking, alcohol drinking, drug taking, unsafe sexual behavior, compensated dating, sex with multiple partners, and gambling in the past three months. Other-destructive delinquency was measured by participation in triadic activities, intimidation, fighting, bullying, gang brawls, theft, trickery, damage to public properties, and stealthy snapping in the past three months. The measurement of these two forms of self-destructive and other-destructive delinquency among at-risk female adolescents was based on recent pertinent studies (Cheung & Yeung, 2017; J. W. K. Yeung & Kim, 2021). If the adolescent girl had taken part in any of these delinquent acts in the forms of self-destructive and other-destructive delinquency, a score of 1 was assigned, and the accumulated count scores hence range from 0 to 9, respectively.

3.2.2. Positive family functioning

In this study, six items were used to measure positive family functioning that characterize quality parent-child relationships, parental concern and care, mutual support, and efficient family communication, which have been used in recent empirical research to measure effective and positive family processes and interactions (Jerf W. K. Yeung, 2021; J. W. K. Yeung et al., 2019). Example items answered for at-risk adolescent girls’ responses include “My mother or father will respect and accept my opinions on important issues” and “It is enjoyable for me to get along with my mother or father”. The items were measured by a 4-point scale from almost not (1) to always (4); higher scores connote better positive family functioning. The alpha coefficient is α = 0.890, indicative of very good reliability.  

3.2.3. Constructive school experiences

For constructive school experiences, nine items modeled from existing research measuring positive teacher-student relationships and effective school engagement were used to tap on the constructive school experiences of at-risk female adolescents (Sethi & Scales, 2020; Woo, Heo, Jang, & Jang, 2023). Example items include “In school, teachers give me adequate opportunities to develop my strengths” and “I enjoy participating in school activities”. The items were measured on a 4-point scale ranging from strongly disagree (1) to strongly agree (4); higher scores connote more positive school experiences. The internal consistency is α = 0.909, indicative of excellent internal reliability.

3.2.4. Susceptibility to deviant peer influence

To capture the susceptibility of deviant peer influence, five items were used to measure at-risk female adolescents’ inclination toward molding and accepting their deviant peers’ behaviors by referring to existing relevant research (Bai, Yao, Duan, Sun, & Niu, 2022; Jiang, 2023; J. W. K. Yeung, Cheung, & Kim, 2020). Example items include “If my friends invite me to take drugs together, I will join” and “If my friends invite me to bully others, I will take part in”. The items were measured on a 4-point scale ranging from strongly disagree (1) to strongly agree (4); higher scores represent being more susceptible to the adverse influence of deviant peers. The alpha coefficient is α = 0.900, implying excellent internal consistency.” (Lines 970-1056)

In addition, a new descriptive table, Table 1 has been added in the part of ‘4. Results’ to describe the demographic characteristics of the at-risk female adolescent participants (lines 1305-1440).

For the contents of the measures, they are now added in the part of ‘Appendix’:

“Appendix

  1. A) Delinquency

1) Self-destructive delinquency

  • Truancy
  • Runaway
  • Smoking
  • Alcohol Drinking
  • Drug Taking
  • Unsafe Sexual Behavior
  • Compensated Dating
  • Sex with Multiple Partners
  • Gambling in The Past Three Months

2) Other-destructive delinquency

  • Triadic Activities
  • Intimidation
  • Fighting
  • Bullying
  • Gang Brawls
  • Theft
  • Trickery
  • Damage to Public Properties
  • Stealthy Snapping

  1. B) Positive Family Functioning

“I accept and respect my father's and mother's opinions regarding important issues.”

“My mother or father will respect and accept my opinions on important issues.”

“It is enjoyable for me to get along with my mother or father.”

“My father or mother creates rooms for me to share my inner feelings.”

“My father or mother gives me adequate care.”

“I tell my father or mother about my problems and troubles.”

  1. C) Constructive School Experiences

“Teachers at school give me opportunities to do what I'm good at.”

“Teachers at school are willing to listen to my feelings and opinions.”

“Teachers at school care about me in aspects apart from the academy, including personal interest and emotion.”

“In school, teachers give me adequate opportunities to develop my strengths.”

“I enjoy participating in school activities.”

“I care about school affairs.”

“I am proud to be part of my school.”

“I cherish the chance to study in school.”
“Going to school is so important, even though it is worthy of accepting things that I dislike at school.”

  1. D) Susceptibility to Deviant Peer Influence

“If friends invite me for truancy, I will consider joining them.”

“If friends invite me to smoke, I will consider joining them.”

“If friends invite me to engage in sex-related activities, I will consider joining them.”

“If my friends invite me to take drugs together, I will consider joining them.”

“If my friends invite me to bully others, I will consider taking part in it.””  (lines 2422-2470)

  1. b) There are some inconsistencies regarding the age range of the participants. On page 4 it is stated that it ranges from 11 to 18 years of age, while on page 6 it is stated that it ranges from 12 to 18 years of age.

Reply: After clairfying, the age range of the at-risk female adoescents is now presented as from 11 to 18 years old.

  1. c) In the section on instruments, the following aspects should be clarified:
  2. The variables included in the delinquency section (section 3.2.1) should be clarified as to where these data are collected. I think I understand that they should be collected from the social workers' records, but it is not clear.

Reply: The two forms of at-risk female adoescents’ delinquency are self-reported data collected from at-risk female adoescents, please refer to the newly information regarding data collection in the parts of ‘3.1 Sample and data source’ (lines 941-959).

  1. Regarding the other measures (sections 3.2.2, 3.2.3, 3.2.4), since they are non-standardized instruments, the authors should include all the items of each of the measures as an appendix so that the study can be replicated. In addition, the description of the measures should specify that they were completed by the adolescents, as well as in what form they were completed (e.g., self-report, as an interview, etc.).

Reply: Now more informatin regarding the academic sources and the contents of the measures are added in the parts of ‘3.2 Measures’ and ‘Appendix’, please refer to lines 970 to 1046, and lines 2403 to 2451.

  1. d) The discussion could be enriched a little. Adding information on other papers that agree or disagree with the results found would greatly enrich the work.

Reply: the parts of ‘5. Discussion’ and ‘6. Conclusion’ have been enhanced sustantially by adding more policy-related suggestions to help at-risk female adolescents and limitations of the current study.

Reviewer 2 Report

Comments and Suggestions for Authors

Abstract: The abstract is clear and well-structured. It provides a good overview of the research paper,

[Suggestion for edit] Line 19: but it could benefit from a bit more detail on the research methodology and implications.

Introduction: The introduction effectively establishes the research's importance, theoretical underpinnings, and the unique focus on at-risk female adolescents. It provides a strong rationale for the study and sets the stage for the subsequent research findings. (No suggestions for edits)

Theorotical Framework: (No suggestions for edits)

Research Method: (No suggestions for edits)

Results: Your findings provide valuable insights into the complex interplay between family dynamics, school experiences, peer influences, and delinquent behaviors among at-risk female adolescents. Your study appears to be meticulously designed and well-conducted, aligning with your preference for high-quality research. (No suggestions for edits)

Discussion: (No suggestions for edits)

Conclusion: (No suggestions for edits)

(Suggestion for edit): Add a section for limitations: Acknowledge any limitations of your study, such as the sample size, potential biases, or constraints in data collection. Addressing limitations demonstrates a realistic view of your research and suggests areas for improvement in future studies.

Author Response

Reviewer 2

Abstract: The abstract is clear and well-structured. It provides a good overview of the research paper,

Reply: Thank you, and for further enhancing the presentation of the Abstract, some revisions hav been made.

[Suggestion for edit] Line 19: but it could benefit from a bit more detail on the research methodology and implications.

Reply: Agree, and this point is same to Reviewer 1, for which more information regarding the procedure of data collection, and the contents of the measures are added in the parts of ‘3.1 Sample and data source’, ‘3.2 Measures’, and ‘Appendix’ of revised manuscript.

In 3.1 Sample and data source, for the data collection procedure it is written:

“The present study was a cooperation project with the Hong Kong Young Women’s Christian Association (HKYWCA) with the aim of investigating the situations and problems faced by local adolescent girls at risk and designing pertinent health and social services to cater for their needs. HKYWCA is one of the oldest and largest social service organizations financially supported by the Hong Kong government, and the organization also manifests itself by emphasizing female and gender-related services. The present study recruited 236 at-risk female adolescents who were the clinical cases of frontline social workers in the organization. The data collection procedures were conducted by first introducing the study purpose and contents to pertinent frontline social workers in the organization who serve female adolescents at risk; and the frontline social workers then helped to contact their at-risk female adolescent clients and seek consent from them and their parental guardians if needed to take part in the study. After obtaining consents from female adolescents at risk and their parental guardians if needed, a questionnaire containing the question items regarding family, school, and peer experiences, cognitive and personality traits, and behavioral performance was given to the consented female adolescents at risk to fill up in the service units of HKYWCA. Generally, the participating at-risk female adolescents needed to take around 20 to 30 minutes to complete the questionnaire. Finally, a sample of 236 at-risk female adolescents was successfully surveyed for data analysis.” (lines 941-959)

For the contents of the measures, more information regarding the academic sources of the measured that are derived from and the measure items are now added in the parts of ‘3.2 Measures’ and ‘Appendix’, which have:

“3.2. Measures

In this study, the measures include self-destructive delinquency and other-structural delinquency of at-risk female adolescents that are treated as the outcome variables, and positive family functioning, constructive school experiences, and deviant peer influence are analyzed as the main predictor variables, along with adjusting for the effects of demographic covariates of at-risk female adolescentsage, educational attainment, current working identity, religious beliefs, and civic organization memberships. The contents of the measurement items for the study variables are listed in the Appendix.

3.2.1. Delinquency

Self-destructive delinquency of female adolescents at risk was measured by participation in the delinquent acts of truancy, runaway, smoking, alcohol drinking, drug taking, unsafe sexual behavior, compensated dating, sex with multiple partners, and gambling in the past three months. Other-destructive delinquency was measured by participation in triadic activities, intimidation, fighting, bullying, gang brawls, theft, trickery, damage to public properties, and stealthy snapping in the past three months. The measurement of these two forms of self-destructive and other-destructive delinquency among at-risk female adolescents was based on recent pertinent studies (Cheung & Yeung, 2017; J. W. K. Yeung & Kim, 2021). If the adolescent girl had taken part in any of these delinquent acts in the forms of self-destructive and other-destructive delinquency, a score of 1 was assigned, and the accumulated count scores hence range from 0 to 9, respectively.

3.2.2. Positive family functioning

In this study, six items were used to measure positive family functioning that characterize quality parent-child relationships, parental concern and care, mutual support, and efficient family communication, which have been used in recent empirical research to measure effective and positive family processes and interactions (Jerf W. K. Yeung, 2021; J. W. K. Yeung et al., 2019). Example items answered for at-risk adolescent girls’ responses include “My mother or father will respect and accept my opinions on important issues” and “It is enjoyable for me to get along with my mother or father”. The items were measured by a 4-point scale from almost not (1) to always (4); higher scores connote better positive family functioning. The alpha coefficient is α = 0.890, indicative of very good reliability.  

3.2.3. Constructive school experiences

For constructive school experiences, nine items modeled from existing research measuring positive teacher-student relationships and effective school engagement were used to tap on the constructive school experiences of at-risk female adolescents (Sethi & Scales, 2020; Woo, Heo, Jang, & Jang, 2023). Example items include “In school, teachers give me adequate opportunities to develop my strengths” and “I enjoy participating in school activities”. The items were measured on a 4-point scale ranging from strongly disagree (1) to strongly agree (4); higher scores connote more positive school experiences. The internal consistency is α = 0.909, indicative of excellent internal reliability.

3.2.4. Susceptibility to deviant peer influence

To capture the susceptibility of deviant peer influence, five items were used to measure at-risk female adolescents’ inclination toward molding and accepting their deviant peers’ behaviors by referring to existing relevant research (Bai, Yao, Duan, Sun, & Niu, 2022; Jiang, 2023; J. W. K. Yeung, Cheung, & Kim, 2020). Example items include “If my friends invite me to take drugs together, I will join” and “If my friends invite me to bully others, I will take part in”. The items were measured on a 4-point scale ranging from strongly disagree (1) to strongly agree (4); higher scores represent being more susceptible to the adverse influence of deviant peers. The alpha coefficient is α = 0.900, implying excellent internal consistency.” (lines 970 -1056)

“Appendix

  1. A) Delinquency

1) Self-destructive delinquency

  • Truancy
  • Runaway
  • Smoking
  • Alcohol Drinking
  • Drug Taking
  • Unsafe Sexual Behavior
  • Compensated Dating
  • Sex with Multiple Partners
  • Gambling in The Past Three Months

2) Other-destructive delinquency

  • Triadic Activities
  • Intimidation
  • Fighting
  • Bullying
  • Gang Brawls
  • Theft
  • Trickery
  • Damage to Public Properties
  • Stealthy Snapping

  1. B) Positive Family Functioning

“I accept and respect my father's and mother's opinions regarding important issues.”

“My mother or father will respect and accept my opinions on important issues.”

“It is enjoyable for me to get along with my mother or father.”

“My father or mother creates rooms for me to share my inner feelings.”

“My father or mother gives me adequate care.”

“I tell my father or mother about my problems and troubles.”

  1. C) Constructive School Experiences

“Teachers at school give me opportunities to do what I'm good at.”

“Teachers at school are willing to listen to my feelings and opinions.”

“Teachers at school care about me in aspects apart from the academy, including personal interest and emotion.”

“In school, teachers give me adequate opportunities to develop my strengths.”

“I enjoy participating in school activities.”

“I care about school affairs.”

“I am proud to be part of my school.”

“I cherish the chance to study in school.”
“Going to school is so important, even though it is worthy of accepting things that I dislike at school.”

  1. D) Susceptibility to Deviant Peer Influence

“If friends invite me for truancy, I will consider joining them.”

“If friends invite me to smoke, I will consider joining them.”

“If friends invite me to engage in sex-related activities, I will consider joining them.”

“If my friends invite me to take drugs together, I will consider joining them.”

“If my friends invite me to bully others, I will consider taking part in it.””  (lines 2422-2470)

For study implications, now the parts of ‘5. Discussion’ and ‘6. Conclusion’ have been revised to include more information for policy implementations and interventions and also future research directions.

Introduction: The introduction effectively establishes the research's importance, theoretical underpinnings, and the unique focus on at-risk female adolescents. It provides a strong rationale for the study and sets the stage for the subsequent research findings. (No suggestions for edits)

Reply: Thank you, and now the part of Introduction has been also enhanced to make the study purpose, values, and contributions of the current study more accurate, clearer, and understandable.

Theorotical Framework: (No suggestions for edits)

Reply: Thank you, and now the part of Theorotical Framework has been also enhanced to make the study purpose, study relationships, and hypotheses more precise and correct.

Research Method: (No suggestions for edits)

Reply: Thank you, and now the part of Research Method has been also enhanced to give more information regarding data collection and sources and contents of the measures are informative and detailed.

Results: Your findings provide valuable insights into the complex interplay between family dynamics, school experiences, peer influences, and delinquent behaviors among at-risk female adolescents. Your study appears to be meticulously designed and well-conducted, aligning with your preference for high-quality research. (No suggestions for edits)

Reply: Thank you, and now descriptive statistics of at-risk female adoelscents’ are given Table 1 in the part of ‘Results’, and other tables are re-checked and revised to elimate the typos.

Discussion: (No suggestions for edits)

Reply: Thank you, and the part of ‘Discussion’ has been enhanced to include more policy- and intervention-related implications for the contributions of the current study.  

Conclusion: (No suggestions for edits)

(Suggestion for edit): Add a section for limitations: Acknowledge any limitations of your study, such as the sample size, potential biases, or constraints in data collection. Addressing limitations demonstrates a realistic view of your research and suggests areas for improvement in future studies.

Reply: The study limitations are now added in the part of ‘Conclusion’, which are written:

“The present study, to the author’s knowledge, was the first attempt to contemporaneously examine the effects of family, school, and peer systems on at-risk female adolescents’ self-destructive and other-destructive delinquency and corroborated that these three crucial social systems are all together predictive of the different forms of delinquency with a different strength among this vulnerable population, which can provide valuable empirical evidence for policy and service innovations and interventions to help female adolescents at risk have better prosocial development by improving their malfunctioning living environment and creating a nurturing socialization context. Future research should explore more other contextual systems concomitantly in relation to youths’ psychological and behavioral development, e.g., neighborhood and religion. Furthermore, a research lens should focus on how these different contextual systems interact with each other and also how different adolescent personality factors, e.g., resilience, align with different contextual systems to contribute to adolescent development. In fact, there are some limitations in the current study that should be rectified in the future. First, the results of this study were only based on a convenient sample of Chinese adolescent girls at risk recruited by a large NGO in Hong Kong. Future research should invite more NGOs in different geographical areas that provide interventions and services for youth to help increase the sample size and sample diversity for enhancing external validity. In addition, cross-nation samples of at-risk female adolescents are suggested for future research, which can help promote generalizability and empirical comparison. Third, the data of the current study was collected from a cross-sectional design, and future research should use longitudinal research data to trace the development and changes of at-risk female adolescents’ psychological and behavioral development and how different contextual systems affect the development and changes. Lastly, a longitudinal design with a multi-informant approach to data collection is suggested for future studies, which not only can help answer the causal validity of the study relationships between different contextual systems and at-risk female adolescents’ different forms of delinquency but also can verify the results from different perspectives to reach a more comprehensive picture of at-risk female adolescent development.” (lines 2191-2220).       

Finally, I would like to give my appreciation to the efforts of Reviewer 1 and Reviewer 2 for their useful suggestions.

Round 2

Reviewer 1 Report

Comments and Suggestions for Authors

After rereading the revised version of the manuscript, I believe that the authors have followed the guidelines of the reviewers and have provided a new manuscript that has greatly improved its quality.